# PeerJ

# Use of Twitter to monitor attitudes toward depression and schizophrenia: an exploratory study

Nicola J. Reavley[1] and Pamela D. Pilkington[2]

[1] Melbourne School of Population and Global Health, University of Melbourne, Victoria, Australia
[2] School of Psychology, Faculty of Health Sciences, Australian Catholic University, Victoria, Australia

Corresponding author
Nicola J. Reavley,
nreavley@unimelb.edu.au

## ABSTRACT

**Introduction.** The paper reports on an exploratory study of the usefulness of Twitter for unobtrusive assessment of stigmatizing attitudes in the community.

**Materials and Methods.** Tweets with the hashtags #depression or #schizophrenia posted on Twitter during a 7-day period were collected. Tweets were categorised based on their content and user information and also on the extent to which they indicated a stigmatising attitude towards depression or schizophrenia (stigmatising, personal experience of stigma, supportive, neutral, or anti-stigma). Tweets that indicated stigmatising attitudes or personal experiences of stigma were further grouped into the following subthemes: social distance, dangerousness, snap out of it, personal weakness, inaccurate beliefs, mocking or trivializing, and self-stigma.

**Results and Discussion.** Tweets on depression mostly related to resources for consumers (34%), or advertised services or products for individuals with depression (20%). The majority of schizophrenia tweets aimed to increase awareness of schizophrenia (29%) or reported on research findings (22%). Tweets on depression were largely supportive (65%) or neutral (27%). A number of tweets were specifically anti-stigma (7%). Less than 1% of tweets reflected stigmatising attitudes (0.7%) or personal experience of stigma (0.1%). More than one third of the tweets which reflected stigmatising attitudes were mocking or trivialising towards individuals with depression (37%). The attitude that individuals with depression should "snap out of it" was evident in 30% of the stigmatising tweets. The majority of tweets relating to schizophrenia were categorised as supportive (42%) or neutral (43%). Almost 10% of tweets were explicitly anti-stigma. The percentage of tweets showing stigmatising attitudes was 5%, while less than 1% of tweets described personal experiences of stigmatising attitudes towards individuals with schizophrenia. Of the tweets that indicated stigmatising attitudes, most reflected inaccurate beliefs about schizophrenia being multiple personality disorder (52%) or mocked or trivialised individuals with schizophrenia (33%).

**Conclusions.** The study supports the use of analysis of Twitter content to unobtrusively measure attitudes towards mental illness, both supportive and stigmatising. The results of the study may be useful in assisting mental health promotion and advocacy organisations to provide information about resources and support, raise awareness and counter common stigmatising attitudes.

## INTRODUCTION

Mental illness affects as many as one in five people in any 12-month period (*Kessler et al., 2007*). Many people with a mental illness experience social and economic hardship as a direct result of their illness. They must cope with their symptoms and also with stigma and discrimination that result from misconceptions about these illnesses (*McNair et al., 2002*; *Corrigan et al., 2003*). These issues are a key concern as stigma may deter people with symptoms from seeking help due to embarrassment or the belief that others will think badly of them (*Gulliver, Griffiths & Christensen, 2012*; *Yap, Reavley & Jorm, 2013*). Stigma may also compound the experience of psychological distress and can adversely affect personal relationships and the ability to achieve educational and vocational goals (*Wells et al., 1994*; *Link et al., 1997*; *Corrigan, 2004*).

In an effort to better understand the factors predicting stigmatising attitudes to mental illness, a number of research studies have focused on media coverage of mental disorders (*Morgan & Jorm, 2009*; *McGinty et al., 2014*). A study of newspaper reporting on mental illness in the USA found that 39% of reports concerned dangerousness and violence (*Corrigan et al., 2005*). There is evidence that such reports can increase stigmatising attitudes, including beliefs about dangerousness. Data from surveys conducted in Germany showed an increase in beliefs about dangerousness after two widely-publicised attacks on politicians by people with schizophrenia (*Angermeyer & Matschinger, 1996*). In another German study, reading a newspaper article about mental illness and violent crime increased stigmatising attitudes (*Dietrich et al., 2006*). However, the majority of studies assessing the role of media coverage of mental illness has focused on traditional media, with relatively little attention paid to newer social media platforms such as Facebook and Twitter (*Egan & Moreno, 2011*; *Moreno et al., 2011*).

Twitter is a popular social networking website that allows users to post brief messages up to 140 characters in length, referred to as tweets. Twitter enables social communication and networking between users through retweeting (reposting other users' tweets to one's own feed), responding to tweets, and following other users' Twitter feeds. Hashtags are included in tweets to identify a theme and enable other users to view related posts with the same hashtag. It is estimated that 18% of online US adults use Twitter (*Pew Internet & American Life Project, 2011*), with similar numbers reported in other developed countries (*Office for National Statistics, 2013*). Unless Twitter users mark their tweets as private, these are public, making Twitter a potentially valuable source of information about the views of people on a range of topics. A number of studies have analysed Twitter content on health-related topics, including an influenza outbreak (*Chew & Eysenbach, 2010*), problem drinking (*West et al., 2012*), dental pain (*Heaivilin et al., 2011*), palliative medicine (*Nwosu et al., 2014*), physical activity (*Zhang et al., 2013*), dementia (*Robillard et al., 2013*), vaccination (*Love et al., 2013*), and breast cancer (*Thackeray et al., 2013*). *Martinez-Perez et al. (2014)* analysed

the purposes and functions of Facebook and Twitter groups for different mental illnesses, including depression. The results showed that individuals with mental illness utilise online social networks to obtain information, access support, and raise awareness. However, to our knowledge, the current study is the first to conduct a detailed analysis of Twitter content as it relates to attitudes to mental illness. The aim of the study is to provide insights into how Twitter users share information about depression and schizophrenia, the type of information shared, and the relative proportions of supportive or stigmatising attitudes in that content. The study offers the opportunity for unobtrusive attitude measurement (*Lee, 2000*), something which is currently lacking in much conventional stigma research, which uses self-report attitude scales that run the risk of social desirability bias because the responses are given to a mental health researcher.

## METHOD

A qualitative analysis was conducted to explore supportive or stigmatising attitudes towards depression and schizophrenia on Twitter.

### Data extraction

The second author retrieved all available English-language tweets that included the hashtags *#depression* or *#schizophrenia* posted on Twitter (http://www.twitter.com) twice daily for a 7-day period between the 27th of November and the 4th of December 2013. Twitter's advanced search function was used to limit the tweets to those in English. The data were retrieved using NCapture for Chrome (*QSR International, 2012a*), a web browser extension that captures web pages, online PDFs, and social media for analysis in NVivo 10 (*QSR International, 2012b*).

Accessing tweets directly through http://www.twitter.com provides access to what is referred to as the Streaming Application Programing Interface (API). Twitter reports that only 1% of existing tweets are available through the Streaming API. Access to all tweets is gained through what is termed the "firehose", which is costly to access, and requires sufficient servers, network availability, and disc space. *Morstatter et al. (2013)* found that more than 40% of tweets in the "firehose" were available to the Streaming API when data were accessed on a daily basis. Therefore, we captured tweets twice daily to maximise our access to the available tweets.

The data retrieved included the tweet and related information (time and date tweeted, number of retweets, hashtags, and mentions), as well as information about the user (username, brief bio, location), and the extent of their Twitter use (number of followers, number following, and total number of tweets). We visually inspected the time and date of each tweet to confirm that all tweets available during the time period were captured continuously. Tweets were included in the analyses if they referred to depression or schizophrenia as mental illnesses. Tweets were excluded based on the following criteria: (a) focused on a topic other than depression or schizophrenia; (b) referred to temporary levels of depressive affect; (c) referred to economic, historical, or meteorological depression (e.g., tropical depression, which is a type of cyclone); (d) could not be accurately coded due to lack of context or the use of hashtags only; or (e) was in a language other than English.

Ethics approval was not obtained as this is not needed for content analysis of publicly available information.

## Content analysis

A content analysis of the tweets was conducted using NVivo 10. The second author coded the tweets based on the extent to which they indicated a particular attitude towards depression or schizophrenia (stigmatising, personal experience of stigma, supportive, neutral, or anti-stigma). Tweets that indicated stigmatising attitudes or personal experiences of stigma were further grouped into the following subthemes: social distance, dangerousness, snap out of it, personal weakness, inaccurate beliefs, mocking or trivializing, and self-stigma. These categories were developed by consensus and are based on previous research investigating stigmatizing attitudes, which has shown that stigmatising attitudes can be grouped into categories, including desire for social distance from the person, beliefs that mental illness is due to personal weakness and that people with mental illness are dangerous (*Link, 1987*; *Griffiths, Christensen & Jorm, 2008*; *Reavley & Jorm, 2012*).

Tweets were also categorised based on their content and user information. The following content themes emerged: (1) personal experience of mental illness, (2) awareness promotion, (3) research findings, (4) resources for consumers (5) advertising, (6) news media, (7) personal opinion or dyadic interaction.

Users were coded for type (individuals, consumer, health professional, organisation, or advocate) and location based on their username and biographic information.

Table 1 provides theme definitions and representative examples of tweets coded within each theme. Both authors coded 5–10% of the tweets and users to ensure accurate coding. Initial levels of agreement ranged between 67–80%. Discrepancies were resolved through discussion and consultation with a third person.

## RESULTS

A total of 7,295 depression tweets and 500 schizophrenia tweets were retrieved from Twitter over the 7-day period. After applying the inclusion and exclusion criteria, 5,907 depression tweets (81%) and 451 schizophrenia tweets (90%) were included in the analyses. NCapture identified between 24 and 29% of the included tweets as retweets. The majority of tweets included links to websites (75%–78%).

## Users

Over the 7-day period, 2,019 users tweeted using the depression hashtag, while 230 users tweeted using the schizophrenia hashtag.

### *Location*

Location data was available for 63% of users. The majority of users who listed their location were from the United States of America (31%), followed by the United Kingdom (15%), Canada (8%), and Australia (3%). The remaining 6% of tweeters were located in Europe, Asia, the Middle East, Africa, South America, and New Zealand.

**Table 1 Theme definitions and representative tweets.**

| Theme | Definition | Representative #depression tweets | Representative #schizophrenia tweets |
|---|---|---|---|
| **Content** | | | |
| **Personal experience of mental illness** | Describes personal experience with depression or schizophrenia as a consumer, or a friend or relative of someone with depression or schizophrenia. | • #depression makes me feel like i'm constantly living a outer body experience when i'm on my own.<br>• Those thoughts again. #depression | • I was diagnosed with dementia praecox (#schizophrenia) 32 years ago. Sorry about my crazy tweets, I'm insane.<br>• 19th Birthday in a psychiatric hospital isn't fun dual diagnosis #autism #schizophrenia http://t.co/KtW43vFmYV |
| **Awareness promotion** | Promotes awareness of depression or schizophrenia by either (1) providing information (2) linking to information on a website (e.g., blog, news article, e-book, or YouTube video) or (3) encouraging discussion about depression or schizophrenia. | • Are You Depressed? Signs and Symptoms of Depression in Women: http://t.co/h2YR0SYZdW #depression Please RT<br>• What does #depression feel like? Mental Health Advocate @ArthurGallant27 offers his perspective here: http://t.co/CTlUFl6dez | • Five myths—and the facts—about schizophrenia http://t.co/yr9WNrwR5r #Stigma #Schizophrenia #MentalHealth #1In4<br>• Check out "Schizophrenia Documentary" on #Vimeo http://t.co/8Zq0TV5ihJ #mentalillness #schizophrenia #depression #breakdown #MentalHealth |
| **Resources for consumers** | Provides resources, advice, or support specifically for consumers or friends and relatives of someone with depression or schizophrenia; or encourages social networking between consumers. | • RT @Daniel_L_Baker: if you're suffering from #depression or know someone who is, download my FREE e-book at http://t.co/oi3n2PobPL<br>• How to Exercise When Depressed—Even if You Prefer Staying in Bed: Want to treat your… http://t.co/tIq1pmn5ft #Bipolar #Depression | • 6 tips for families dealing with #schizophrenia http://t.co/pIBFQGYTCs<br>• Information for young people recovering from psychosis—This booklet was designed for… http://t.co/ziN5gAtNxr #bcss #schizophrenia |
| **News media** | Describes and links to an online news story or article | • Toronto woman denied entry to U.S. after agent cites history of #depression http://t.co/lHu9GPgRjl via @torontostar<br>• BBC News—Call for faster patient access to talking therapies http://t.co/XWzR9ol4iW #depression #mentalhealth #anxiety #therapy | • @HuffingtonPost article on #Veterans battle with #schizophrenia http://t.co/sCQqA5EayQ<br>• #Schizophrenia sufferers' 'job fears' http://t.co/lSGpnTo1Mb #ukmh |

Table 1 (*continued*)

| Theme | Definition | Representative #depression tweets | Representative #schizophrenia tweets |
|---|---|---|---|
| **Advertising** | Advertises a product or service for sale | ● #Frankincense—Deep in History—Treat many medical conditions— #Depression—#Anxiety #Cancer & more! #PleaseVisit http://t.co/WIr1Q219iX ● RT @SkypeTherapist: Online Cognitive Therapy (CBT) for #Depression. Talk to a Mindfulness therapist via Skype. http://t.co/ykzYPLfpyH. Emai… | ● Haloperidol schizophrenia best #schizophrenia website licensed drug store. http://t.co/FJdLH56709 #rogers ● Size Order #schizophrenia Female-rx-plus-oil Online, To Use http://t.co/KOYuVXmAQ5 #tiquanunderwood |
| **Research findings** | Describes the outcomes of a research study | ● #Depression in #pregnancy: new study shows preferences for #therapy over #medication http://t.co/XOzpgDNu00 ● Study finds low #zinc associated with #depression. See: http://t.co/wHks3KEr5p | ● Meta-analysis finds that mindfulness may help with the negative symptoms of #schizophrenia http://t.co/xRRNIgJ7wK #Mindfulness ● Links Between #Schizophrenia & Cardiovascular Disease— http://t.co/tPbSy6XSQV |
| **Personal opinion or dyadic interaction** | Describes an individual's personal opinion or view | ● I think this #generation has issues with #depression because everyone thinks about themselves too much. What happened to putting others 1st? ● #Suicide is a #symptom of an #illness and should be treated as such. #depression | ● The Schofield's are a family that make me feel grateful for my daughter #January #schizophrenia ● Most assume too quickly, judge inconsiderately, and understand inadequately. http://t.co/tp0xG9ZYXc #Schizophrenia #NotSoUncommon |
| **Attitude Stigmatising** | Indicates a negative attitude towards people with depression/schizophrenia | ● Wanna know the secret of #depression? People only have it if they know they've become a f**ing #loser, the rest is just an #excuse #stfu ● #depression #f**life #crazy #killyourself #crazy #whore #retard #idiot #psycho #horrible #dumb #stupid #bitch #emo http://t.co/aUr63r5ax8 | ● Schizophrenia: Medicine's mystery— Society's shame #schizophrenia http://t.co/BGIVJuSe2J ● Rihanna gets along with the voices in her head #schizophrenia #helprequired |

**Peer**J

Table 1 (*continued*)

| Theme | Definition | Representative #depression tweets | Representative #schizophrenia tweets |
|---|---|---|---|
| **Personal experience of stigma** | Describes a personal experience of stigmatising attitudes or discrimination towards depression/schizophrenia | • My husband suffered from #depression & #anxiety. I was amazed on how many said he should "just get over it". It's a disease, not a choice.<br>• #quote #patience #depression pic, twitter, comtRB98Lbake, People make assumptions about us and that really gets to me", | • @seniorchuffy @dROP2RIPPLE @andrewnorthcbc people with SMI left out shows that there's little engagement w/people with #Schizophrenia here<br>• #MentalHealth #Schizophrenia Many people are not on my side most take advantage of me and many are still out to… http://t.co/3yS8IvS61U |
| **Anti-stigma** | Explicitly promotes the reduction of stigma towards people with depression/schizophrenia | • RT @Daniel_L_Baker: people can't just "snap out of #depression". It doesn't work like that. If it did, don't you think they would?<br>• Are you sick of being belittled for having #depression? Stand up for yourself with these 5 comebacks: http://t.co/2uZxiaFh96 | • And if you think that people with #schizophrenia (like me) or other mental health conditions are "psycho" or dangerous, shame on you.<br>• Stand up to the #mentalhealth stigma: http://t.co/Nz0WpGgspc<br>• #Depression #Bipolar #Schizophrenia #Anxiety #PTSD |
| **Supportive** | Supportive towards people with depression/schizophrenia | • It IS possible to recover from #depression. Never give up: http://t.co/sQ8Ihu0pkd<br>• @Oprah yes!!!! If we could all be a little kinder, there would be less suffering, lower rates of #depression, and improved #mentalhealth | • @SchizophreniaCa—I support my son who has #schizophrenia. My husband supports me #love |
| **Neutral** | Indicates a neutral attitude towards people with depression/ schizophrenia | • Our paper on prevalence of #depression and #BipolarDisorder within UK Biobank, just published @PLOSONE http://t.co/6nFj…<br>• Sometimes you just hit a point where life just feels like it's too hard and you just want to give up. #depression #fml | • Symptoms of #schizophrenia are very common, raising questions about how distinct the diagnosis is. http://t.co/6yZSOto8dK #m…<br>• Just submitted my third essay!! #relax #psychology #schizophrenia |

(*continued on next page*)

Table 1 (*continued*)

| Theme | Definition | Representative #depression tweets | Representative #schizophrenia tweets |
|---|---|---|---|
| **Type of stigma social distance** | Unwillingness to have social contact with people with depression/schizophrenia | [a] | • The Schofield's are a family that make me feel grateful for my daughter #January #schizophrenia<br>• Seeing these poor babies with these illnesses makes me want to hold my boys and thank god that they are healthy#elhamdallah #schizophrenia |
| **Dangerousness** | Belief that people with depression/schizophrenia are dangerous | [a] | • Untreated #Schizophrenia Boosts Likelihood of Future Violence Among Prisoners http://t.co/vWlmtU6CQd |
| **Snap out of it** | Belief that people with depression/schizophrenia could snap out of it | • How low can a person go b4 I can scold the person to wake up get a job get a life? #depression #stubborn #pride preventing accepting help | [a] |
| **Personal weakness** | Belief that depression/schizophrenia is a sign of personal weakness | • #depression is a crisis of confidence http://t.co/q3jumRiTZ9 | [a] |
| **Inaccurate beliefs** | Indicates a lack of knowledge or inaccurate understanding of depression/schizophrenia is. E.g., That schizophrenia refers to having multiple personalities. | | • Btw, Katie is my other personality #schizophrenia<br>• Getting ready to hang out with myself #schizophrenia |
| **Mocking or trivialising** | Rude, derogatory, or trivialising towards people with depression/schizophrenia | • Tyson: To get rid of depression you sit in a cupboard and pee your pants! Creager: Nooooooo! #depression | • @richardrainey you realize that it looks like we are having a twitter conversation with ourselves. #schizophrenia |
| **Self-stigma** | Indicates that the consumer has internalised a stigmatising attitude towards people with depression/schizophrenia | • Wish it wasn't so hard to ask for help #depression #wordsiwillneversay #Truth | [a] |

**Notes.**

[a] No tweets identified for this theme.

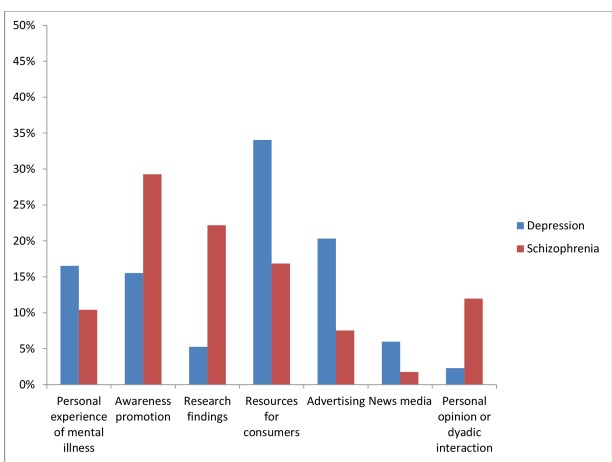

**Figure 1** Content of tweets by mental illness.

### User type

Users tweeting about depression were mostly consumers (35%), followed by individuals (unspecified) (27%), organisations (25%), health professionals (9%), and mental health advocates (3%). In contrast, the majority of users tweeting about schizophrenia were individuals (54%), followed by organisations (25%), health professionals (8%), mental health advocates (8%), and consumers (4%).

## Content

The content of the tweets by mental illness is presented in Fig. 1.

*Depression:* Tweets on depression mostly related to resources for consumers (34%; 2,011 tweets) or advertised services or products for individuals with depression (20%; 1,201 tweets). The remaining tweets described personal experiences with depression (16%; 977 tweets), promoted awareness of depression (15%; 917 tweets), linked to news media articles on depression (6%; 354 tweets), reported research findings (5%; 311 tweets), or described an individual's personal opinion or view (2%; 136 tweets). The hashtags which most commonly accompanied the hashtag *#depression* were *#anxiety*, *#mentalhealth*, *#bipolar*, *#stress*, and *#suicide*.

*Schizophrenia:* The majority of schizophrenia tweets aimed to increase awareness of schizophrenia (29%; 132 tweets) or reported on research findings (22%; 100 tweets). Tweets providing resources, advice, or support for people with schizophrenia accounted for 17% of the tweets (76 tweets). Twelve percent of tweets described personal opinions (54 tweets), while 10% described personal experiences with schizophrenia (47 tweets). A small number of tweets consisted of advertising (7%; 34 tweets) and news media articles (2% 8 tweets). The hashtags which most commonly accompanied the hashtag *#schizophrenia* were *#mental*, *#health*, *#mentalhealth*, *#bcss*, *#psych*. The hashtag *#bcss* refers to the British Columbia Schizophrenia Society.

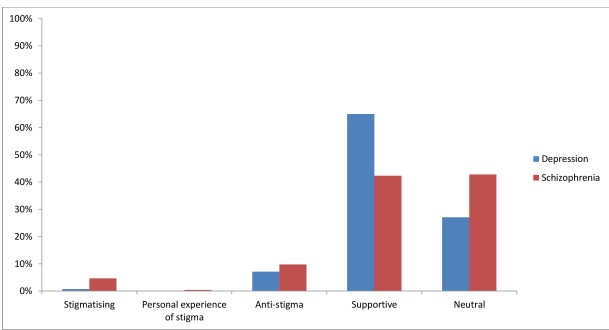

**Figure 2  Types of attitudes by mental illness.**

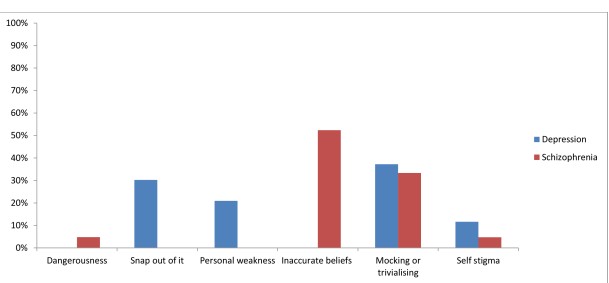

**Figure 3  Types of stigmatising attitudes by mental illness.**

### Attitudes towards depression or schizophrenia

The numbers of stigmatising attitudes by mental illness is shown in Fig. 2. Figure 3 presents the types of stigmatising attitudes that were evident.

*Depression:* Tweets on depression were largely supportive (65%; 3,839 tweets) or neutral (27%; 1,600 tweets). A number of tweets were specifically anti-stigma (7%; 421 tweets). Less than 1% of tweets reflected stigmatising attitudes (0.7%; 43 tweets) or personal experience of stigma (<0.1%; 4 tweets). More than one third of the tweets which reflected stigmatising attitudes were mocking or trivialising towards individuals with depression (37%; 16 tweets). The attitude that individuals with depression should "snap out of it" was evident in 30% (13 tweets) of the stigmatising tweets. The belief that people with depression are weak was displayed in 21% (9 tweets) of tweets. A further 12% (5 tweets) of the tweets suggested self-stigma.

*Schizophrenia:* The majority of tweets relating to schizophrenia were categorised as neutral (43%; 193 tweets) or supportive (42%; 191 tweets). Almost 10% of tweets were explicitly anti-stigma (44 tweets). The percentage of stigmatising attitudes was 5% (21 tweets), while less than 1% of tweets described personal experiences of stigmatising attitudes towards individuals with schizophrenia (2 tweets). Of the tweets that indicated stigmatising attitudes, most reflected inaccurate beliefs about schizophrenia being multiple personality disorder (52%; 11 tweets) or mocked or trivialised individuals with schizophrenia (33%; 7 tweets). The remaining tweets indicated an unwillingness to be in social contact with

people with schizophrenia (5%; 1 tweet), a belief that individuals with schizophrenia are dangerous (5%; 1 tweet), or were suggestive of self-stigma (5%; 1 tweet).

# DISCUSSION

Content analysis of Twitter information about depression and schizophrenia can provide insights into the type of information shared, who is sharing the information, and the types of attitudes displayed, whether supportive or stigmatising. An analysis of tweets with the hashtags *#depression* or *#schizophrenia* posted on Twitter during a 7-day period revealed that most tweets for both illnesses were supportive or neutral, although stigmatising attitudes were more prevalent in the schizophrenia tweets. More than one third of the depression tweets which reflected stigmatising attitudes were mocking or trivialising towards individuals with depression and the belief that individuals with depression should "snap out of it" was evident in almost one third. To some extent, these findings reflect surveys assessing stigmatising attitudes towards people with depression, which have shown the beliefs that people are weak rather than sick are relatively common (*Reavley & Jorm, 2011*), although views about the dangerousness of people with depression, which are relatively common in surveys were not reflected in the tweets. Of the schizophrenia tweets that indicated stigmatising attitudes, just over half reflected inaccurate beliefs about schizophrenia being multiple personality disorder while one third mocked or trivialised individuals with schizophrenia.

In terms of the type of information, the results of the current study showed that tweets on depression mostly related to resources for consumers or advertised services or products for individuals with depression, whereas the majority of schizophrenia tweets aimed to increase awareness of schizophrenia or reported on research findings. These differences are likely to relate to the higher prevalence of depression than schizophrenia and therefore the potentially greater market for products and services. These findings may be compared to those of a recent study by *Martinez-Perez et al. (2014)*, who analysed the purposes and functions of Facebook and Twitter groups for different mental illnesses, including depression. They classified groups according to whether they were support groups, self-help groups, advocacy and awareness groups and fundraising groups. They concluded that self-help groups were the most common category (64%), followed by support groups (15%), and advocacy and awareness groups (10%). They also identified the creators of the groups, listing the following categories: associations, societies, foundations, collectives, individuals, companies, caregivers, and specialists. Over 70% of creators were identified as individuals. However, they did not do further analysis on the content of the tweets.

As far as we are aware, this is the first study to assess the content of tweets on the topic of mental illness. Strengths of the study include the opportunity for unobtrusive measurement of freely volunteered views (*Lee, 2000*). Results of such studies are less likely to be affected by social desirability bias than responses to surveys, which are more commonly used to assess stigmatising attitudes (*Angermeyer & Matschinger, 2004*; *Reavley & Jorm, 2011*).

With the increasing prominence of social media, the results of the study may be useful in assisting mental health promotion and advocacy organisations to provide information about resources and support, raise awareness and reduce stigma. Twitter may also be an increasingly useful resource for people seeking health information on the internet as a number of tweets provided links to health information. There is evidence that the quality of online information is improving, particularly that relating to affective disorders, and it is likely that online information and interventions will continue to become more important (*Reavley & Jorm, 2010*). In addition, advocacy organisations may want to specifically counter common stigmatising attitudes such as beliefs about weakness in people with depression or beliefs about 'split personality' in people with schizophrenia. A recent campaign to raise awareness and counter the use of stigmatising terms for people with intellectual disabilities has taken the approach of directly responding to celebrities who use these terms (*Sibley, 2014*).

The main limitation of the study is our inability to determine what proportion of existing tweets referring to depression and anxiety were analysed. Comparisons between the Streaming API and the "firehose" suggest that the Streaming API may not sufficiently represent the activity on Twitter as a whole (*Morstatter et al., 2013*). Despite this, we believe that it was appropriate to access the Streaming API as the significant costs involved in accessing the "firehose" would not yield proportionate benefits. We analysed a sufficient number of tweets to reach data saturation. Furthermore, the Streaming API is available to the public, and therefore representative of how the average person experiences Twitter.

A second potential limitation is our focus on tweets using the depression and schizophrenia hashtags. This may have meant that content related to other hashtags or slang terms such as "schizo" (which people with stigmatising attitudes are potentially more likely to use) may have been missed. Moreover, the anonymity of most users makes it impossible to verify the authenticity of content or to assess stigmatising attitudes according to age, gender, or location. Finally, these findings cannot be generalised to communities outside Twitter.

Given the speed of reactions to events on social media, it would be useful for future research to compare attitudes before and after significant events, for which it is not possible to anticipate a 'pre-test' (e.g., a mass shooting that involves a person with a mental illness) as this may also provide useful information on stigmatising attitudes.

## CONCLUSION

The study supports the use of analysis of Twitter content to unobtrusively measure attitudes towards mental illness, both supportive and stigmatising. The results of the study may be useful in assisting mental health promotion and advocacy organisations to provide information about resources and support, raise awareness and counter common stigmatising attitudes.

### Funding

The authors received salary support from the National Health and Medical Research Council. The funders had no role in study design, data collection and analysis, decision to publish, or preparation of the manuscript.

### Competing Interests

The authors declare there are no competing interests.

### Author Contributions

- Nicola J. Reavley conceived and designed the experiments, performed the experiments, contributed reagents/materials/analysis tools, wrote the paper, reviewed drafts of the paper.
- Pamela D. Pilkington conceived and designed the experiments, analyzed the data, contributed reagents/materials/analysis tools, wrote the paper, prepared figures and/or tables, reviewed drafts of the paper.

### Supplemental Information

Supplemental information for this article can be found online at http://dx.doi.org/10.7717/peerj.647#supplemental-information.

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
