# Peer review of "Use of Twitter to monitor attitudes toward depression and schizophrenia: an exploratory study"

_PeerJ, doi:10.7717/peerj.647_

## Round 0.1 · original submission · Major Revisions

Dear Author,There are numerous major issues which this paper has to resolve before it can be resubmitted which is: This paper does not specify if the researchers accessed the full stream of Twitter data (“firehose” data) via a licensed company or if they used the 1% of Twitter data that is publicly available. I have to assume that the researchers only accessed 1% of the Twitter data since these methods are not detailed in the paper. This is a very substantial limitation of the current study that should have been acknowledged. In addition, it is difficult to assess if this study was conducted rigorously and to a high technical standard without sufficient details on how the Twitter data was collected.

The researchers focus their paper on the monitoring of stigmatizing attitudes towards depression and schizophrenia. This approach is problematic for a number of reasons. First, there are so few Tweets that are identified as representing stigmatizing attitudes towards depression (0.7%) and schizophrenia (5%). More rationale is needed to support a paper that focuses on the content analysis of fewer than 50 Tweets posted over a 7-day span. This is especially the case given that over 500 million Tweets are streamed daily. Second, according to the findings, the most prevalent theme of the depression and schizophrenia Tweets are “supportive” messages towards these mental illnesses. This finding implies that Twitter messages which alleviate stigmatizing attitudes towards mental illness are a predominant topic, especially versus Tweets that present stigmatizing messages. The relevance and meaningfulness of this study are in question in light of such findings.

Another is In methods, "A qualitative analysis was conducted to establish the prevalence of stigmatising attitudes towards depression and schizophrenia on Twitter" : COMMENT: Because hashtag depression and hashtag schizophrenia were only used, we cannot know if the data collection was comprehensive for all tweets related to depression and schizophrenia on Twitter, which you discuss in your limitations. This particular limitation does not allow us to calculate prevalence so this cannot be a goal of the present study.

Please indicate how the different categories for content analysis were determined.Did one researcher categorize the tweets or were the different categories developed by consensus? Are the categories comprehensive to your sample, meaning do all the tweets in the sample fall into at least one category?

Unique Users: Were some of these tweets tweeted over and over again by one user? Please discuss the limitations of including repeat users.

Validity of the findings

It should be pointed out that we cannot generalize these findings to communities outside of Twitter.

Furthermore, prevalence cannot accurately be assessed without knowing you’ve developed a list of search terms that will completely capture all tweets related to depression and schizophrenia.

Until these experimental problems are solved / can be done there may be major issues in the next round of re-review which will be undertaken by the same peer reviewers.

·

Basic reporting

A new look at an old problem. I think the author deserve credit for using social media in scientific study. Also agreed that this is the first study to assess the content of tweets on mental illness.

Experimental design

Exploratory research design by using social media has limitations as highlighted in the discussion part. However it is not clear whether this type of research design requires consent and permission or whether author has gotten consent and permission from participants.
Also noticed familiar and unfamiliar names, i.e., Rihanna, Schofield's. Is mentioning name considered unethical or may this has legal implication?

Validity of the findings

I suggest that the finding were compared with similar study that using conventional method, i.e., pen and paper

Additional comments

No comment

Reviewer 2 ·

Basic reporting

I found this paper difficult to follow, especially with regards to the number of Tweets that were ultimately included in all of the analysis. The authors do not provide any sample size numbers, only percentages, throughout their paper so the exact numbers of Tweets that were coded is unclear. These n’s should be explicitly reported in the Results section and Tables.

Experimental design

This paper does not specify if the researchers accessed the full stream of Twitter data (“firehose” data) via a licensed company or if they used the 1% of Twitter data that is publically available. I have to assume that the researchers only accessed 1% of the Twitter data since these methods are not detailed in the paper. This is a very substantial limitation of the current study that should have been acknowledged. In addition, it is difficult to assess if this study was conducted rigorously and to a high technical standard without sufficient details on how the Twitter data was collected.

The researchers focus their paper on the monitoring of stigmatizing attitudes towards depression and schizophrenia. This approach is problematic for a number of reasons. First, there are so few Tweets that are identified as representing stigmatizing attitudes towards depression (0.7%) and schizophrenia (5%). More rationale is needed to support a paper that focuses on the content analysis of fewer than 50 Tweets posted over a 7-day span. This is especially the case given that over 500 million Tweets are streamed daily. Second, according to the findings, the most prevalent theme of the depression and schizophrenia Tweets are “supportive” messages towards these mental illnesses. This finding implies that Twitter messages which alleviate stigmatizing attitudes towards mental illness are a predominant topic, especially versus Tweets that present stigmatizing messages. The relevance and meaningfulness of this study are in question in light of such findings.

Validity of the findings

The conclusions are not supported by the primary findings of this study. It appears to me that the findings of this study contradict much of what is presented in the Discussion section. The Discussion did not acknowledge or apply the finding that most of the depression and schizophrenia Tweets were “supportive”. Instead, the primary implications presented in the Discussion section emphasize a need to counter the stigmatizing attitudes on Twitter. The researchers also encourage health professionals to increase awareness about depression and schizophrenia by providing internet links to health information websites, presumably on Twitter. These implications are in direct contrast with the actual findings of this study which indicate that Tweets on depression and schizophrenia are, for the most part, “largely supportive” of these mental illnesses. In addition, most of the Tweets in this study (nearly 80%) do include links to websites which also contradicts the researchers’ recommendations to increase the amount of Tweets that provide internet links to health information about mental illnesses. As a whole, the researchers are providing implications that are not supported by their findings. They need to more strongly acknowledge their findings that very few of the Tweets they analyzed actually reflected stigmatizing attitudes and such attitudes are being countered on Twitter.

Additional comments

This study summarizes an exploratory study on the usefulness of Twitter to study stigmatizing attitudes towards depression and schizophrenia. There are a number of significant methodological and interpretative issues described below which reduce the enthusiasm for an otherwise timely study.

·

Basic reporting

Insert Abstract Heading.

In the introduction, "It is estimated that 18% of online adults use Twitter (Pew Internet & American Life Project 2011)". Please specify if this applies to American adults or adults worldwide.

Human Research: If you went through an approval process, please state somewhere in the document. Or, if there was not an approval process, please state reasoning for being exempt.

Experimental design

In methods, "A qualitative analysis was conducted to establish the prevalence of stigmatising attitudes towards depression and schizophrenia on Twitter" : COMMENT: Because hashtag depression and hashtag schizophrenia were only used, we cannot know if the data collection was comprehensive for all tweets related to depression and schizophrenia on Twitter, which you discuss in your limitations. This particular limitation does not allow us to calculate prevalence so this cannot be a goal of the present study.

Please indicate how the different categories for content analysis were determined.Did one researcher categorize the tweets or were the different categories developed by consensus? Are the categories comprehensive to your sample, meaning do all the tweets in the sample fall into at least one category?

Unique Users: Were some of these tweets tweeted over and over again by one user? Please discuss the limitations of including repeat users.

Validity of the findings

It should be pointed out that we cannot generalize these findings to communities outside of Twitter.

Furthermore, prevalence cannot accurately be assessed without knowing you’ve developed a list of search terms that will completely capture all tweets related to depression and schizophrenia.

Additional comments

Interesting study and findings. Please elaborate on limitations (based on comments above) and results in greater detail in Discussion section. Thank you.

---

## Round 0.2 · accepted · Accept

Congratulations.Your manuscript is accepted for publication in Peer J.Thank you for your submission.

·

Basic reporting

I maintained my recommendation that the paper is acceptable for publication and furthermore the author has responded and made necessary amendments as suggested in the first review.

Experimental design

no comment

Validity of the findings

no comment

Additional comments

I maintained my recommendation that the paper is acceptable for publication and furthermore the author has responded and made necessary amendments as suggested in the first review.